

# The sicariid spiders in the state of Bahia, Brazil (Arachnida: Araneae)

Júlia Andrade-de-Sá[1], Tania Kobler Brazil[1], Rejâne Maria Lira-da-Silva[1] and Antonio Domingos Brescovit[2]

[1] Institute of Biology, Federal University of Bahia, Salvador, Bahia, Brazil
[2] Zoological Collections Laboratory, Instituto Butantan, São Paulo, São Paulo, Brazil

## ABSTRACT

**Background**. Sicariidae is a small family of cryptic and recluse spiders, with 178 species grouped into three genera: *Hexophtalma*, *Loxosceles* and *Sicarius*. Only the last two occur in Brazil, where 29 species are recorded, with a greater number of records in the Southeast and South regions. To date, there is no regionalized study of the distribution of these spiders in the Northeast of Brazil. We presented more than four decades of data on the distribution of sicariid spiders in the state of Bahia. Our research aimed to identify and evaluate the distribution of sicariid species in this region, characterizing them in their biomes and phytophysiognomies.

**Methods**. Data covers a period from 1983 to 2024, obtained from articles describing species recorded in Bahia and from three scientific collections from Brazilian institutions: Universidade Federal da Bahia (MHNBA); Instituto Butantan (IBSP); Universidade Federal de Minas Gerais (UFMG). Maps were produced using the QGIS 3.34.1 software, based on geographic coordinates obtained from the original literature, when provided. Species without information on geographic coordinates were georeferenced using the *SpeciesLink* (geoLoc tool) for the municipalities. Biomes were consulted on the georeference Brazilian institute website and the phytophysiognomies were ignored when not specified in the data of collections or original literature.

**Results**. We confirmed 14 species of sicariid spiders in the state of Bahia, eight from the genus Loxosceles (i) and six from *Sicarius*. The species were recorded in 48 municipalities, located mainly in the Caatinga biome. We recorded a distribution expansion of seven species: *Loxosceles amazonica, L. chapadensis, L. karstica, L. similis, Sicarius cariri, S. ornatus* and *S. tropicus*. There is a difference in species composition in the two regions separated by the São Francisco River, an important geographic barrier especially for small invertebrates. More than 75% of the species were registered in the east side and about 20% exclusively in the west side: *L. boqueirao, L. cardosoi, L. carinhanha*. This suggests the need for future studies to evaluate additional variables that determine community structure, especially considering their low vagility and reclusive behavior. Although *Sicarius* are found almost exclusively in the Caatinga, there are isolated records from Restinga and dry forest enclaves in the Cerrado domain and Atlantic Forest areas. This reinforces the possibility of them being able to adapt to slightly wetter environments erase colonize and survive in seasonally dry rainforest areas. Unlike Sicarius (i), Loxosceles (i) occupy a wide variety of habitats in natural, urban and domiciliary situations. Many of the records here such as for *Loxosceles boqueirao, L. cardosoi, L. carinhanha, L. chapadensis, L. karstica* and *L. troglobia* were

Corresponding author
Júlia Andrade-de-Sá,
jubs.andrade.2014@gmail.com

obtained at caves or nearby areas, which appear to be attractive habitats for these nocturnal spiders.

# INTRODUCTION

There are still few publications on the distribution of spiders in Brazilian territory, not only due to its continental extension but also due to the scarcity of inventories, financial support, and taxonomist arachnologists, especially in the Northeast, North, and Midwest regions of Brazil (*Marques & Lamas, 2006*). Of the 52,240 species of spiders spread across the globe (*World Spider Catalog, 2024*) more than 3,000 occur in Brazil (*Brescovit, Oliveira & Santos, 2011*) and around 400 species are recorded in the state of Bahia (*Lira-da-Silva, 2011*).

The Sicariidae family comprises 178 species grouped into three genera: *Hexophthalma* Karsch, 1879 ($n = 8$), distributed in Africa, *Loxosceles* Heineken & Lowe, 1832 ($n = 149$) and *Sicarius* Walckenaer, 1847 ($n = 21$), both with an emphasis on Neotropical Region (*World Spider Catalog, 2024*). In Brazil, 29 species of this family are registered (*World Spider Catalog, 2024*). For the state of Bahia, the latest checklist records four species: *Loxosceles amazonica Gertsch, 1967*, *L. chapadensis Bertani, Fukushima & Nagahama, 2010*, *L. similis* Moenkhaus, 1898 and *Sicarius tropicus* (*Mello-Leitão, 1936*) in *Lira-da-Silva (2011)*.

The articles published by *Magalhães, Brescovit & Santos (2013)* and *Magalhães, Brescovit & Santos (2017)* can be considered those that offer the most complete and current data on *Sicarius*, including a phylogenetic review. The authors present the description of four new species of *Sicarius* occurring in Bahia: *S. cariri, S. diadorim, S. ornatus* and *S. saci*. In addition to these records, we can highlight works with specific occurrences for the state involving the descriptions of *Loxosceles boqueirao Bertani et al., 2024*; *L. chapadensis Bertani, Fukushima & Nagahama, 2010*; *L. carinhanha, L. karstica Bertani et al., 2018* and *L. troglobia Souza & Ferreira, 2018*.

Sicariid spiders are haplogynous with only six eyes arranged in dyads (*Magalhães, Brescovit & Santos, 2017*) and all have withdrawn behavior. Both *Hexophthalma* (African) and *Sicarius* (Neotropical) are part of the Sicariinae subfamily largely restricted to dry environments, namely deserts, xeric shrublands, and dry forests (*Magalhães, Brescovit & Santos, 2017*).

The genus *Sicarius* occurs in xeric environments in South and Central America, mostly in deserts and seasonally dry tropical forests. They do not spin webs and have a unique behavior of burying themselves, covering their bodies with grains of sand from the substrate, due to the small thorns that cover them (*Magalhães, Brescovit & Santos, 2013*; *Magalhães, Brescovit & Santos, 2017*).

Out of the 21 valid species for the South American continent, seven occur in Brazil (*World Spider Catalog, 2024*), and five in Bahia (*Magalhães, Brescovit & Santos, 2013*,

*2017*). Despite being phylogenetically in a different clade from *Loxosceles* (subfamily Loxoscelinae), they possess the same enzyme in their venom (phospholipase D) responsible for the dermonecrotic activity observed in loxoscelism accidents (*Arán-Sekul et al., 2020*). Studies on the action of *Sicarius* venom and envenomation are still limited and only a few species have been characterized regarding their toxic potential (*Lopes et al., 2013*; *Lopes et al., 2021a*; *Lopes et al., 2021b*).

*Loxosceles* spiders are well known for their medical significance and have a widespread global distribution, with 22 species occurring in Brazil (*World Spider Catalog, 2024*). Envenomation by *Loxosceles* is characterized by local lesions, with erythema, edema, inflammation, and dermonecrosis as well as systemic reactions such as hemolysis, renal failure, and hematologic alterations that include thrombocytopenia, disseminated intravascular coagulation, and hemolytic anemia (*Da Silva et al. 2004*; *Swanson & Vetter, 2006*; *Chaim et al., 2011*).

The Brazilian Ministry of Health recorded approximately 38,000 accidents caused by these spiders in the period from 2019 to 2023, the majority in the Southeast and South regions (*Ministry of Health of Brazil, 2024*). However, these recorded numbers probably do not reflect the real number of accidents caused by *Loxosceles*. Cases can be underreported, as there is often no development of symptoms and no reference to spider bites by the victim, or overreported, as many of the differential symptomatic characteristics of loxoscelism are confused with several other diseases (*Vetter, 2015*). This attribution of clinical effects to various spiders is problematic due to vague case definitions and a lack of clinical evidence (*Isbister & White, 2004*; *Vetter, 2015*). However, in Brazil there are significant challenges in training the hospital emergency teams, who typically handle accident cases, to identify spiders accurately at any taxonomic level. As a consequence, during the period from 2019 to 2023, the majority of cases (63,9%) were simply reported as ''spiders'' (*Ministry of Health of Brazil, 2024*).

They construct irregular webs in crevices of cliffs, under tree bark, tiles, and bricks, behind frames and furniture, and in clothing, typically seeking shelter from light. They are not aggressive and bite mainly when pressed against the body. Accidents predominantly occur in adults, where the bites have been recorded on the torso and proximal regions of the limbs (*Ministry of Health of Brazil, 2011*).

The first works that record the occurrence of *Loxosceles* and *Sicarius* in Bahia, except for taxonomic ones, are those published by *Lucas (2009)*, *Barbaro & Cardoso (2009)*, *Brazil et al. (2009)* and *Lira-da-Silva (2011)* and encompass four species, *Loxosceles amazonica*, *L. similis*, *L. chapadensis* and *Sicarius tropicus*. Here we present the compilation of more than four decades of data, aiming to update the distribution of Sicariidae in the state of Bahia. Our main objective was to fill gaps in knowledge about the regional distribution of these species, gathering existing records in literature and scientific collections, including these data in distribution maps, and presenting their relationship with some environmental variables.
## MATERIALS & METHODS

### Species data

The list of species of the Sicariidae from the state of Bahia was obtained from:

#### 1. Scientific collections

The specimens are deposited in the following taxonomic collections (curators indicated between parentheses):

- MHNBA - Museu de História Natural da Bahia, Universidade Federal da Bahia, Salvador (T.K. Brazil) ($n = 257$ registers);
- IBSP–Instituto Butantan, São Paulo (A.D. Brescovit) ($n = 196$ registers);
- UFMG–Universidade Federal de Minas Gerais (A.J. Santos) ($n = 30$ registers).

All MHNBA and UFMG records are available on the SpeciesLink platform (https://specieslink.net/), under the acronym UFBA-ARA (Arachnological Collection of the Museum of Natural History of Bahia–Araneae) and UFMG-ARA (Arachnid Collection of the Taxonomic Collections of UFMG).

The list of species was organized following the respective sub-families and genera, prepared according to this structure: (i) valid species name with author and the year of publication; (ii) historical records from description articles; (iii) new records; (iv) remarks.

### Environment data

#### 1. Locations/Municipalities

Geographical coordinates were obtained from the original literature (when provided). Species without any information on geographic coordinates were georeferenced using the *SpeciesLink* geoLoc tool (http://splink.cria.org.br/geoloc) for the municipalities.

#### 2. Average annual temperature and rainfall

Climate-data.org (https://pt.climate-data.org).

#### 3. Biomes and physiognomies

The biomes were identified for each municipality according to IBGE[1] (https://www.ibge.gov.br/): Caatinga, Cerrado, Atlantic Forest.

The physiognomies were classified according to the description article.

### Maps

The distribution maps were produced using the QGIS 3.34.1 software (*QGIS Development Team, 2024*). Hydrography shapes were obtained from ANA (National Water Agency) Metadata Catalog; Topography shape from ESRI Satellite 2023 World Topographic Basemap (QGIS add-on); Brazil Biomas 2019, South America 2021 and BR/BA UF 2022 from IBGE maps portal.

All data regarding collections, locations/municipalities, coordinates, average annual temperature and rainfall, biomes, and physiognomies have been compiled into a table available in Table S1.

### Biomes of the state of Bahia

The state of Bahia is located in northeastern Brazil and is the 5th largest federal unit in the country, with an area of 564,760.429 km$^2$ (IBGE). It encompasses three

[1] IBGE: Instituto Brasileiro de Geografia e Estatística (Brazilian Institute of Geography and Statistics).

biomes—Caatinga, Cerrado (Neotropical savanna), and Atlantic Forest, each of them with various phytophysiognomies, intrinsically related to its biodiversity. The Restinga (coastal xeric vegetation that grows in sandy soils) for example, is a typical phytophysiognomie from the coastal area of the Atlantic Forest. Caatinga covers the majority (>50%) of Bahia's territory, concentrated in the central region of the state, situated between the coastal Atlantic Forest to the east and the Cerrado to the west. It is the only biome restricted to Brazilian territory, primarily occupying the Northeast region, with some areas in the State of Minas Gerais (*Leal, Tabarelli & Silva, 2003*). Within it is the ecoregion of Chapada Diamantina, which experiences higher humidity due to rainfall, allowing for the development of larger and denser forests compared to other regions of the Caatinga. In some places, there are rock outcrops that ecologically function as deserts, where succulent plants like cacti and bromeliads thrive (*Silva et al., 2017*). In the central region is the Chapada Diamantina National Park (PARNA-CD), located 425 km from the state capital, Salvador, and 1,135 km from the federal capital, Brasília. It encompasses the municipalities of Lençóis, Andaraí, Itaetê, Mucugê, Ibicoara, and Palmeiras, excluding the municipal headquarters. It presents a typically tropical climate, with rainfall ranging between 750 mm and 1,000 mm annually, with 4 to 6 months of dry weather. The terrain is quite rugged, with plateaus, steep ridges, and mountains, usually along the park's margins. The average altitude is around 1,000 m, and the highest point in the park is approximately 1,700 m above sea level. The vegetation consists of rocky fields, open fields, savanna, forests, and patches of vegetation (*Linhares, 2007*). The rainwater network flows into the São Francisco basin on one side, and within the region itself on the other, where the two largest rivers in Bahia originate: the Contas and the Paraguaçu rivers (*Conservation Units in Brazi, 2023*).

The São Francisco River crosses the entire state of Bahia from south to north and serves as a significant geographical barrier, especially for less mobile animals, such as spiders. With a length of 2,863 km, most of the river's sections are located in the semi-arid region of Bahia, part of an area known as the drought polygon, where significant drought periods are recorded (*São Francisco River Basin Committee (CBHSF), 2023*).

**Research Area (Fig. 1)**

# RESULTS

**Sicariidae Keyserling, 1880**

**Distribution of Sicariidae (Fig. 2)**

**Sicariinae** *Magalhães, Brescovit & Santos, 2017*
*Sicarius* **Walckenaer, 1847**

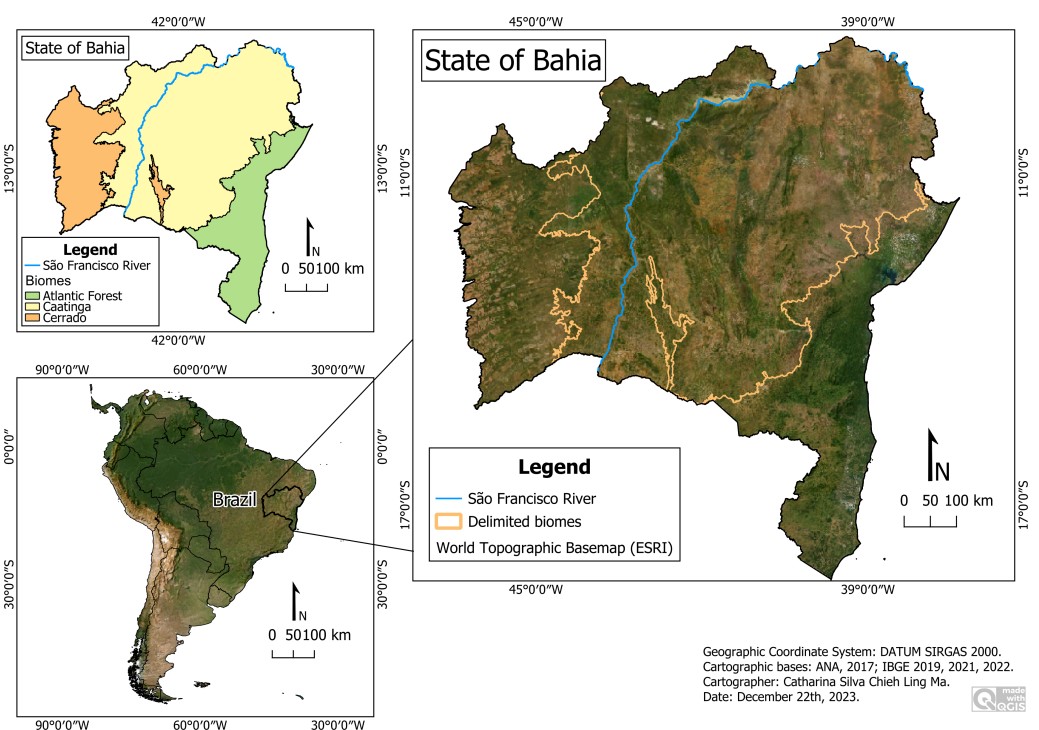

**Figure 1** **Research area.** Map data: ESRI Satellite 2023 World Topographic Basemap.

### *Sicarius cariri* Magalhães, Brescovit & Santos, 2013

**Historical records** BRAZIL. **Bahia:** *Campo Formoso*, Gruta da Boa Vista (40°51′35″W, 10°9′45″S), Gruta Morrinho (40°55′5″W, 10°12′33″S); *Canudos* (39°01′35″W, 9°53′48″S); *Central* (42°6′46″W, 11°8′8″S); Fazenda Paulo (42°6′46″W, 11°8′8″S); Peixe-Candeia (42°6′46″W, 11°8′8″S); Roça do Paulo (42°6′46″W, 11°8′8″S); Toca da Esperança (39°01′35″W, 9°53′48″S); *Curaça*, Riacho Seco (39°54′33″W, 8°59′26″S); *Itaguaçu da Bahia* (42°23′58″W, 11°0′42″S); *Jaborandi*, border with Posse/state of Goiás (46°0′12″W, 13°55′54″S); *Jaguarari* (40°17′59″W, 10°14′16″S); *Miguel Calmon*, Distrito de Bagres (40°35′40.4″W, 11°19′31.8″S); *Paripiranga* (37°55′2″W, 10°37′19″S); *Paulo Afonso*, Estação Ecológica Raso da Catarina (38°28′15.1″W, 9°39′58.1″S); *Queimadas* (39°37′28″W, 10°58′40″S); *Salvador*, Lagoa do Abaeté (38°30′39″W, 12°58′17″S); *Santa Terezinha* (39°31′22″W, 12°46′19″S); *Senhor do Bonfim*, Campus VII da UNEB (40°10′W, 10°26′S) (*Magalhães, Brescovit & Santos, 2013*).

**New records:** BRAZIL. **Bahia**: *Cachoeira*, Rio da Prata, Fazenda Caimbongo (38°57′21″W, 12°37′06″S) (UFBA-ARA 4672); *Caetité* (42°29′07.3″W, 14°03′54.2″S) (UFBA-ARA 3603; 3604; 3605; 4605); *Campo Formoso,* Toca do Angico (40°19′16″W, 10°30′27″S) (IBSP 210113); Gruta Lapa do Convento (40°19′16″W, 10°30′27″S) (IBSP 260652); *Central,* Abrigo Waldemar II (42°6′46″W, 11°8′8″S) (IBSP 13409); *Gentio do Ouro* (42°30′09″W, 11°25′45″S) (UFBA-ARA 5291; 5292; 5293; 5294; 5296; 5297; 5322; 5323; 5324); *Lajedinho* (40°54′20″W, 12°21′21″S) (IBSP 182319; 182324); *Una* (39°04′31″W, 15°17′36″S) (IBSP 27793).

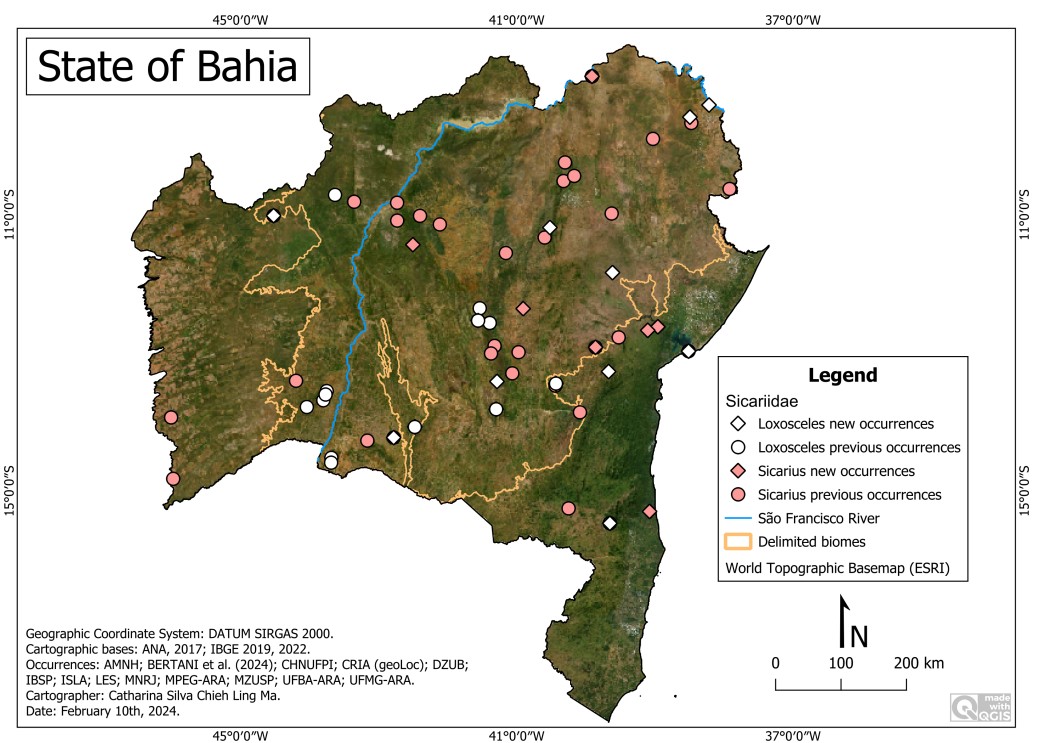

**Figure 2** **Distribution of Sicariidae species in Bahia.** Map data: ESRI Satellite 2023 World Topographic Basemap.

**Remarks:** *Sicarius cariri* was the species with the greatest distribution range among all sicariid species (Fig. 3). It was recorded in 19 municipalities, in all biomes of the State, and in the most varied phytophysiognomies. It has been found in natural environments, including caves and urban environments. Consequently, they occurred in environments with greater climatic variation, from altitudes of 5 m (Cachoeira) to 1,098 m (Gentio do Ouro), with average temperatures of 22 °C (Caetité) to 29.4 °C (Canudos) and precipitation average of 294 mm/year (Canudos) to 1,432 mm/year (Una). We highlight its specific presence in a small environmental protection area (APA) Lagoas e Dunas do Abaeté, an enclave of restinga located in a highly urbanized area of the city of Salvador.

### *Sicarius diadorim* Magalhães, Brescovit & Santos, 2013

**Historical records:** BRAZIL. **Bahia:** *Cocos*, Fazenda Trijunção (45°58′23″W, 14°49′15″S); *Guanambi*, Pátio of Aeroporto (42°44′W, 14°12′S); Ceraíma, 15 Km SW (42°41′16.3″W, 14°17′2.4″S); Fazenda do Fabiano (42°43′56.4″W, 14°10′17.6″S); *Jaborandi*, Fazenda Jatobá (46°0′12″W, 13°55′54″S); *Palmas do Monte Alto*, Serra do Monte Alto (43°09′43″W, 14°16′03″S); *São Félix do Coribe* (44°11′40″W, 13°24′03″S) (*Magalhães, Brescovit & Santos, 2013*).

**New records:** None.

**Remarks:** *Sicarius diadorim* was recorded in the Caatinga and Cerrado biomes (Fig. 4). It is recorded for the first time in an urban environment within the limits of the municipality

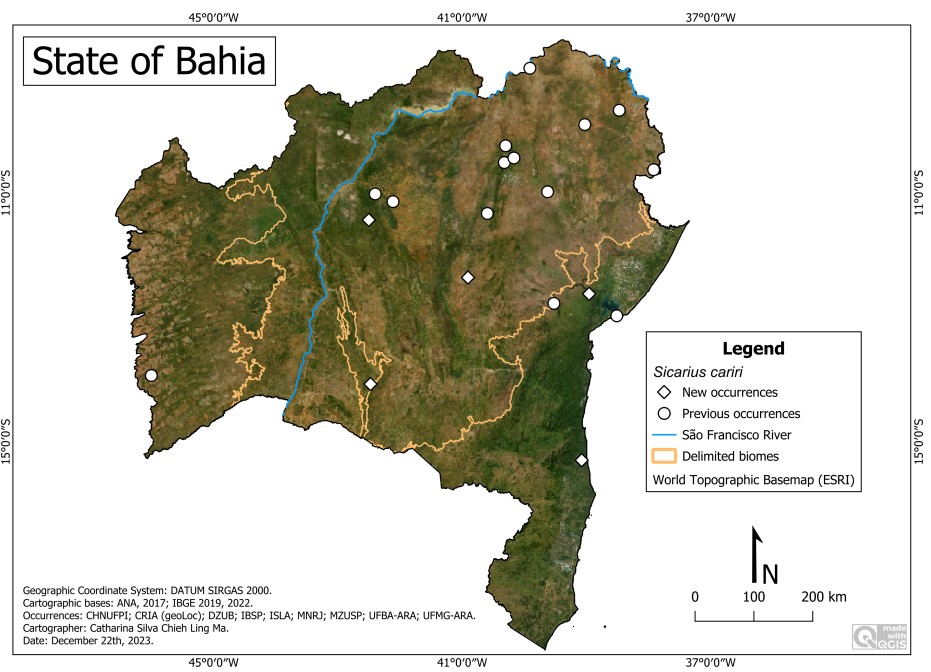

**Figure 3** *Sicarius cariri* **distribution in Bahia.** Map data: ESRI Satellite 2023 World Topographic Basemap (QGIS add-on).

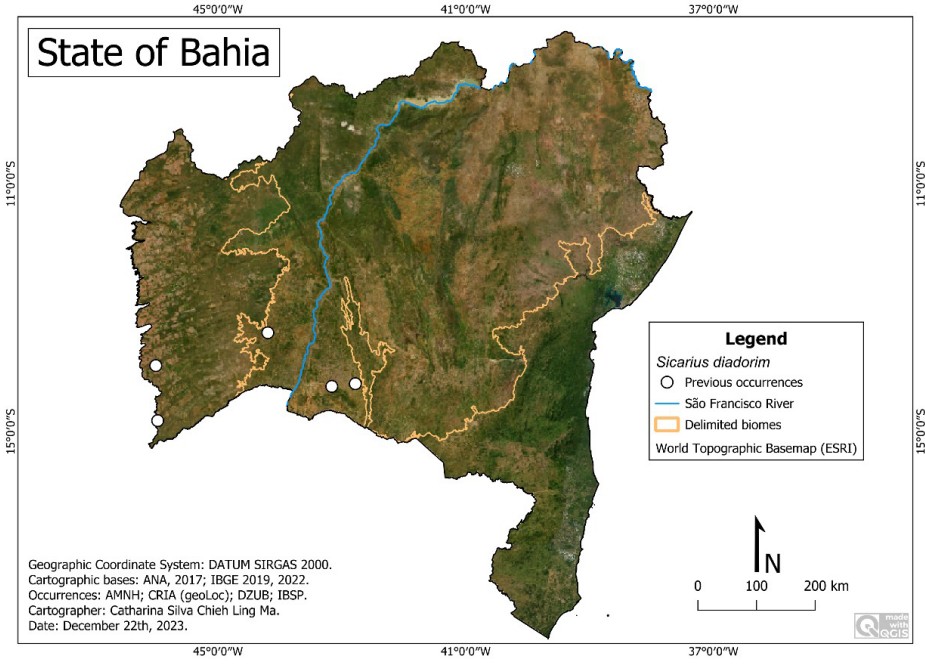

**Figure 4** **Distribution of** *Sicarius diadorim* **in Bahia.** Map data: ESRI Satellite 2023 World Topographic Basemap.

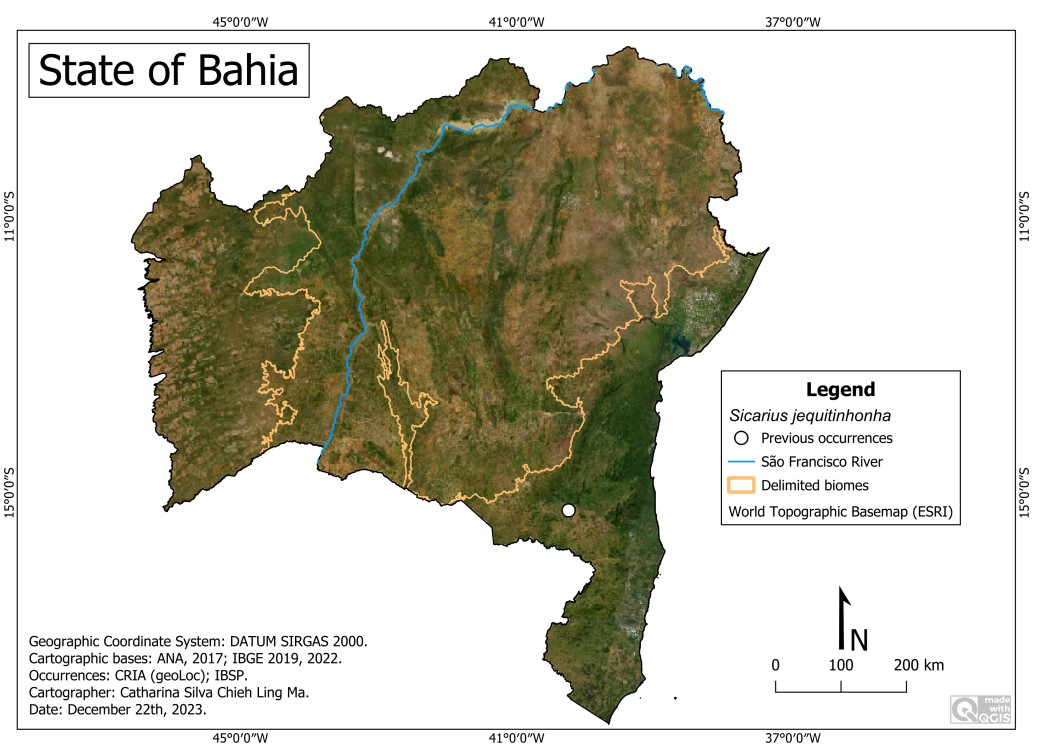

**Figure 5** **Distribution of *Sicarius jequitinhonha* in Bahia.** Map data: ESRI Satellite 2023 World Topographic Basemap.

of Guanambi. It occurs at altitudes between 40 m (São Félix do Coribé) and 802 m (Palmas do Monte Alto), with average temperatures of 25.0 °C (Jaborandi) to 26.0 °C (São Félix do Coribé) and average precipitation of 491 mm/year (Guanambi) to 826 mm/year (Cocos).

### *Sicarius jequitinhonha* Magalhães, Brescovit & Santos, 2017

**Historical records:** BRAZIL. **Bahia:** *Itapetinga* (40°14′52″W, 15°14′57″S) (*Magalhães, Brescovit & Santos, 2017*).

**New records.** None.

**Remarks:** In Bahia, its occurrence was restricted to the municipality of Itapetinga (Fig. 5), south of the state, in the Atlantic Forest, where the altitude is 279 m, average temperature of 23.2 °C and rainfall of 816 millimeters/year.

### *Sicarius ornatus* Magalhães, Brescovit & Santos, 2013

**Historical records:** BRAZIL. **Bahia:** *Andaraí*, Igatu (41°19′10″W, 12°53′44″S); *Iramaia*, Gruta do Calixto (41°3′48″W, 13°17′35″S); *Itaetê*, surroundings of the Gruta Natal, Fazenda Rio Alegre (40°58′20″W, 12°59′12″S); *Ituaçu*, Gruta Mangabeira (41°17′47″W, 13°48′48″S); *Jequié* (40°05′00″W, 13°51′28″S); *Maracás*, close to the headquarters of Ferbasa (40°26′17″W, 13°28′16″S); *Milagres* (39°51′27.9″W, 12°54′54.2″S); *Mucugê*, Reserva Particular do Patrimônio Natural Adília Paraguaçu Batista (41°22′15″W, 13°00′19″S) (*Magalhães, Brescovit & Santos, 2013*).

**New record:** BRAZIL. **Bahia:** *Cachoeira* (38°57′21″W, 12°37′06″S) (UFBA-ARA 4671).

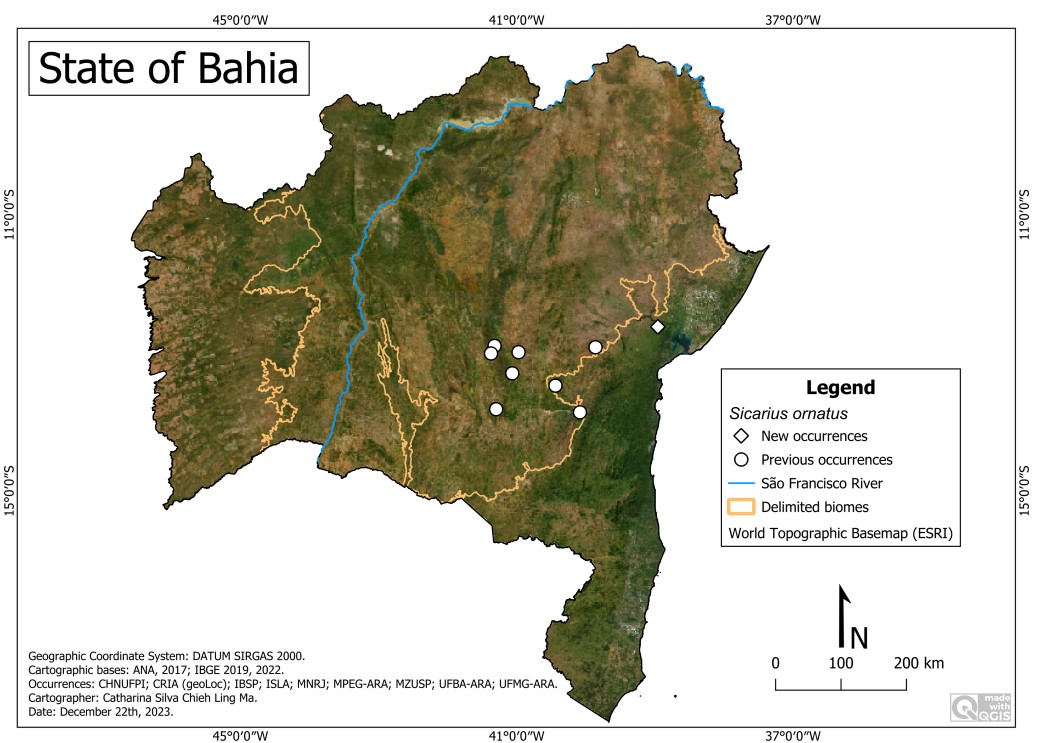

**Figure 6** **Distribution of *Sicarius ornatus* in Bahia.** Map data: ESRI Satellite 2023 World Topographic Basemap.

**Remarks:** *Sicarius ornatus* was distributed in Caatinga and Atlantic Forest environments (Fig. 6). It occurred at altitudes between 5 m (Cachoeira) and 983 m (Mucugê), average temperature of 19.7 °C (Mucugê) to 24.1 °C (Cachoeira) and average precipitation of 464 mm/year (Milagres) to 1,017 (Mucugê) mm/year. We highlight its presence in PARNA-CD (Mucugê), a place with a recognized tourist influx.

### *Sicarius saci* Magalhães, Brescovit & Santos, 2017

**Historical records:** BRAZIL. **Bahia:** *Barra*, Brejo Olhos Dágua (43°21′9.7″W, 10°48′26.8″S); Vila de Ibiraba, São Francisco dunes (43°21′10″W, 10°48′27″S); *Lençóis* (41°23′23″W, 12°33′48″S); *Morro do Chapéu*, Casto (41°22′7,9″W, 11°28′39,2″S); Parque Estadual Morro do Chapéu (41°19′49.3″W, 11°29′51.2′S); *Xique-Xique* (42°43′51″W, 10°49′19″S) (*Magalhães, Brescovit & Santos, 2013*).

 **New records:** None.

 **Remarks:** *Sicarius saci* was distributed only in the Caatinga (Fig. 7). It occurs at altitudes ranging between 394 (Lençóis) and 1,134 m (Morro do Chapéu), average temperature of 20.7 °C (Morro do Chapéu) to 27.4 °C (Barra) and average rainfall of 484 (Xique-Xique) to 637 (Lençóis) millimeters per year. We highlight its presence in PARNA-CD (Lençóis), in a place with a recognized tourist influx.

### *Sicarius tropicus* (Mello-Leitão, 1936)
**Historical records:** None.

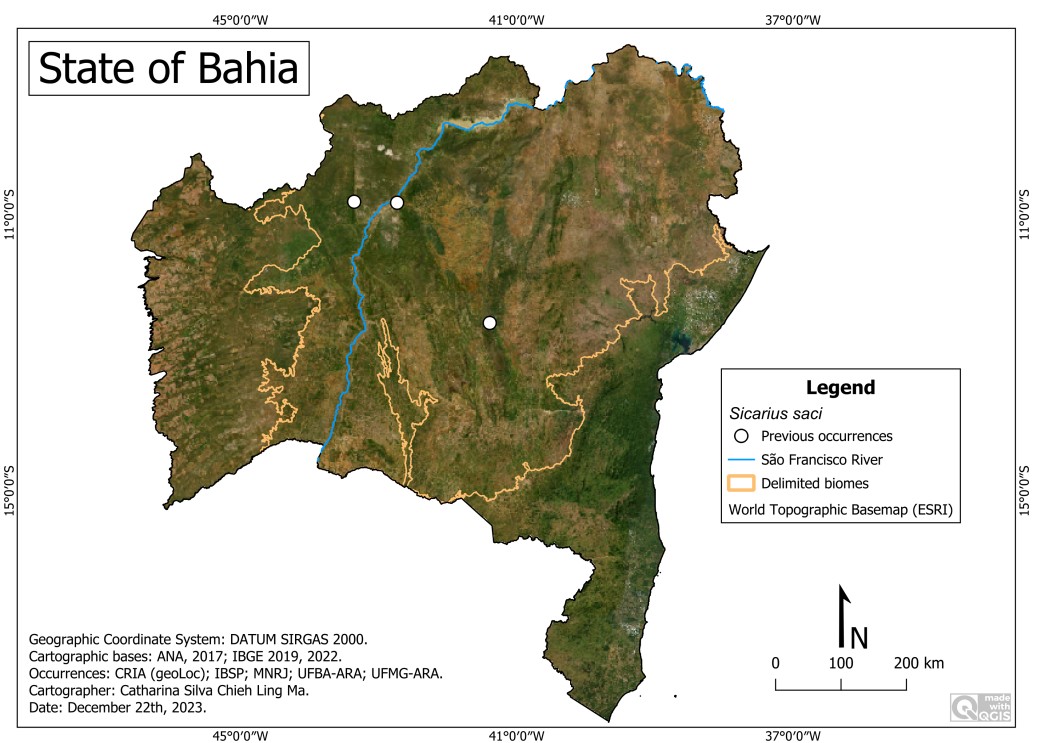

**Figure 7 Distribution of *Sicarius saci* in Bahia.** Map data: ESRI Satellite 2023 World Topographic Basemap.

**New records:** BRAZIL. **Bahia:** *Caetité* (42°28′29″W, 14°04′10″S) (UFBA-ARA 3606); *Curaça*, Riacho Seco (39°54′33″W, 8°59′26″S) (IBSP 161057, 161058, 161059, 161061, 161062, 161064, 161065, 161066, 161067, 161068, 161069, 161070, 161071, 161072, 161073, 161074, 161075, 161083, 161084); *Cruz das Almas*, Recôncavo Baiano (39°06′06″W, 12°40′13″S) (IBSP 86511); *Milagres* (39°51′27,9″W, 12°54′54,2″S) (IBSP 161086, 161087, 161090, 161095, 161097).

**Remarks:** This is the first record of this species for the state of Bahia (Fig. 8). It was previously recorded in the Northeast region in the states of Paraíba and Pernambuco (*Mello-Leitão, 1936*; *Magalhães, Brescovit & Santos, 2013*; *Magalhães, Brescovit & Santos, 2017*). *S. tropicus* is present in municipalities located within areas that encompass all the biomes of the state, occurring at altitudes between 220 (Cruz das Almas) and 824 m (Caetité), average temperature of 22 °C (Caetité) to 26.7 °C (Curaçá) and average precipitation of 413 mm/ year (Curaça) to 1,161 millimeters/year (Cruz das Almas).

## *Sicarius* specimens from the MHNBA (Fig. 9)

Loxoscelinae *Magalhães, Brescovit & Santos, 2017*
*Loxosceles* Heineken & Lowe, 1832

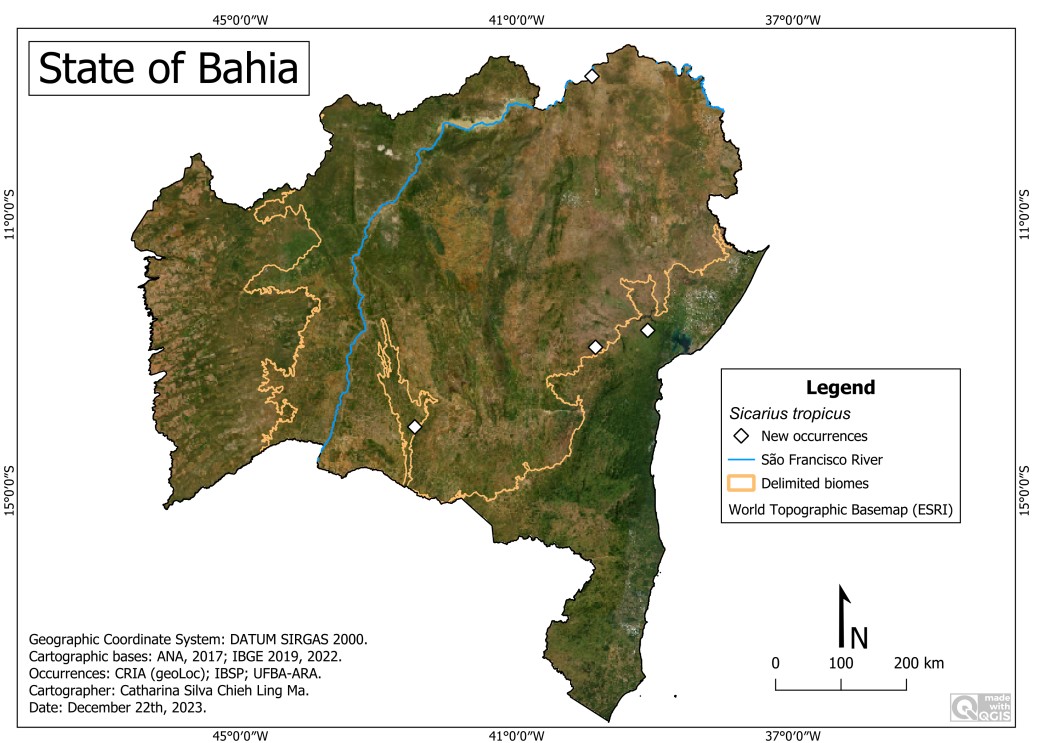

**Figure 8  Distribution of *Sicarius tropicus* in Bahia.** Map data: ESRI Satellite 2023 World Topographic Basemap.

### *Loxosceles amazonica* Gertsch, 1967

**Historical records:** BRAZIL. **Bahia:** *Santa Rita de Cássia* (44°31′09″W, 11°00′31″S); *Buritirama* (43°37′49″W, 10°42′29″S) (*Brazil et al., 2009*).

New records: BRAZIL. **Bahia:** *Paulo Afonso*, Estação Ecológica Raso da Catarina (38°29′29″W, 9°34′53″S) (IBSP-165656); *Salvador,* Lagoa do Abaeté (38°30′39″W, 12°58′17″S) (IBSP 10331).

Remarks: The new records show its occurrence in the Caatinga and Atlantic Forest biome in dunes/restinga (Salvador) (Fig. 10). It occurred at altitudes varying between 8 (Salvador) and 492 m (Buritirama), average temperature of 25.6 °C (Salvador) to 26.8 °C (Santa Rita de Cássia) to and average precipitation of 367 (Paulo Afonso) to 1,235 (Salvador) millimeters annually. It is important to highlight its presence in the APA Lagoas e Dunas do Abaeté, an enclave located in a highly urbanized area of the municipality of Salvador. Although the records of Buritirama and Santa Rita de Cassia were not obtained from either a description article or collections, we chose to include it because it had been previously recorded by the authors (*Brazil et al., 2009*).

### *Loxosceles boqueirao* Bertani & Gallão, 2024

**Historical records:** BRAZIL. **Bahia:** *Carinhanha,* Lapa do Boqueirão Cave (44°02′18″W, 13°46′51″S) (*Bertani et al., 2024*).

New records. None.

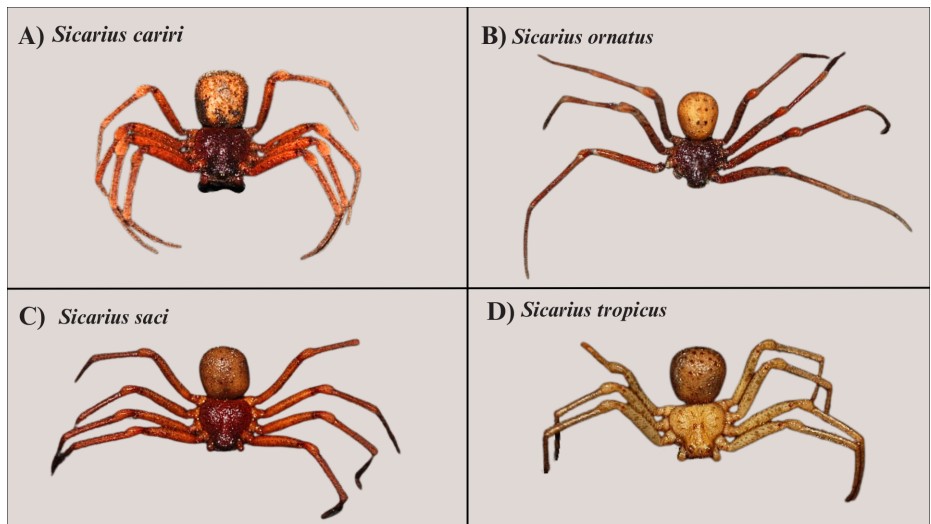

**Figure 9** *Sicarius* **from the MHNBA.** (A) *Sicarius cariri* specimen (UFBA-ARA 5294), female from Gentio do Ouro, Bahia. (B) *S. ornatus* specimen (UFBA-ARA 4671), male from Cachoeira, Bahia.(C) *S. saci* specimen (UFBA-ARA 5224), female from Barra.(D) *S. tropicus* specimen (UFBA-ARA 3606), female from Caetité, Bahia. Photo credit: Júlia Andrade-de-Sá.

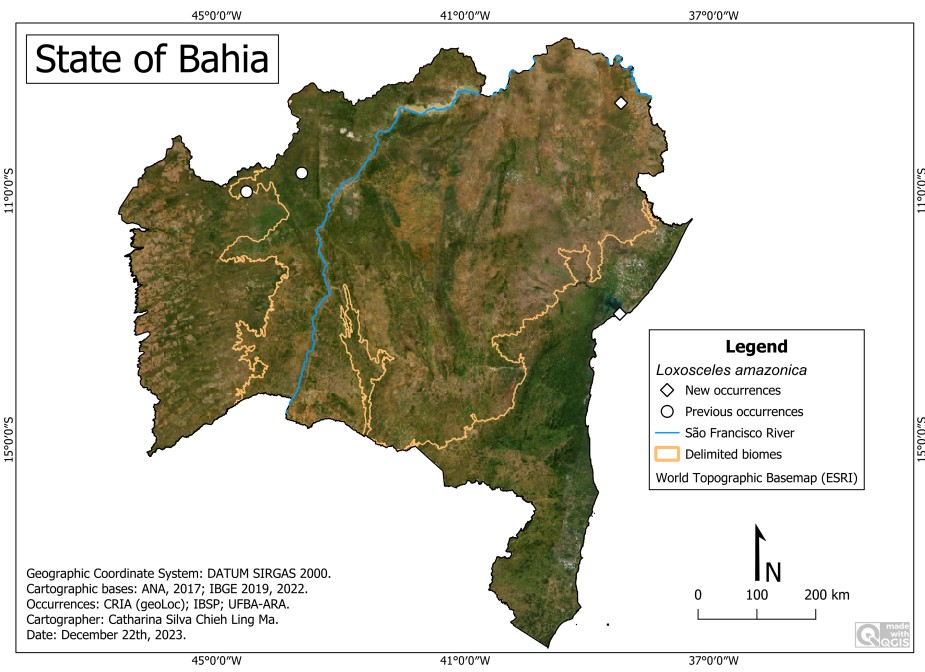

**Figure 10** **Distribution of** *Loxosceles amazonica* **in Bahia.** Map data: ESRI Satellite 2023 World Topographic Basemap.

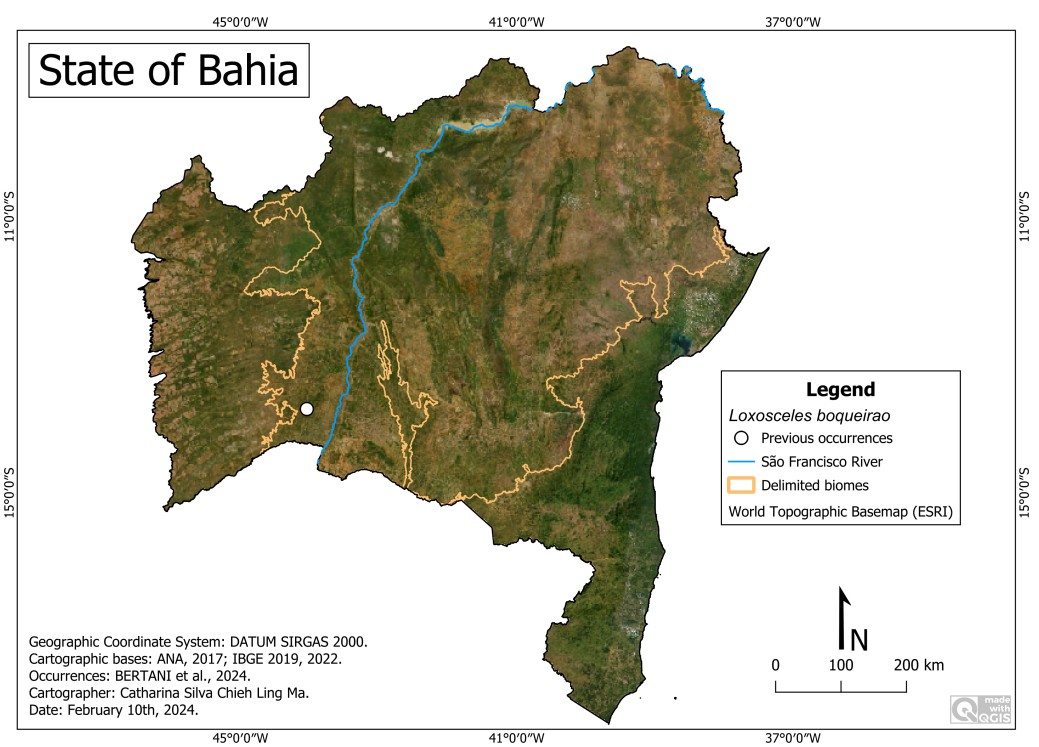

**Figure 11** **Distribution of *Loxosceles boqueirao*.** *Loxosceles boqueirao* distribution in Bahia. Map data: ESRI Satellite 2023 World Topographic Basemap (QGIS add-on).

**Remarks:** All records of this species are restricted to a single cave in the municipality of Carinhanha, in the Caatinga area (Fig. 11). It occurs at an altitude of around 496 m, with an average temperature of 26.3 °C and an average rainfall of 665 millimeters per year.

### *Loxosceles cardosoi* Bertani, von Schimonsky & Gallão, 2018

**Historical records:** BRAZIL. **Bahia:** *Carinhanha,* Gruna da Altina cave (43°45′W, 13°33′S) (*Bertani et al., 2018*).

**New records**. None.

**Remarks:** All records of this species are restricted to a single cave in the municipality of Carinhanha, in the Caatinga area (Fig. 12). It occurs at an altitude of around 496 m, with an average temperature of 26.3 °C and an average rainfall of 665 millimeters per year.

### *Loxosceles carinhanha* Bertani, von Schimonsky & Gallão, 2018

**Historical records:** BRAZIL. **Bahia:** *Carinhanha*, Gruna Água Fina cave (43°48′W, 13°41′S) (*Bertani et al., 2018*).

**New records**. None.

**Remarks:** As with *Loxosceles cardosoi*, all records of *L. carinhanha* are from a cave restricted to the Caatinga of the city of Carinhanha (Fig. 13). It occurs at an altitude of around 484 m, with an average temperature of 26.3 °C and an average rainfall of 665 millimeters per year.

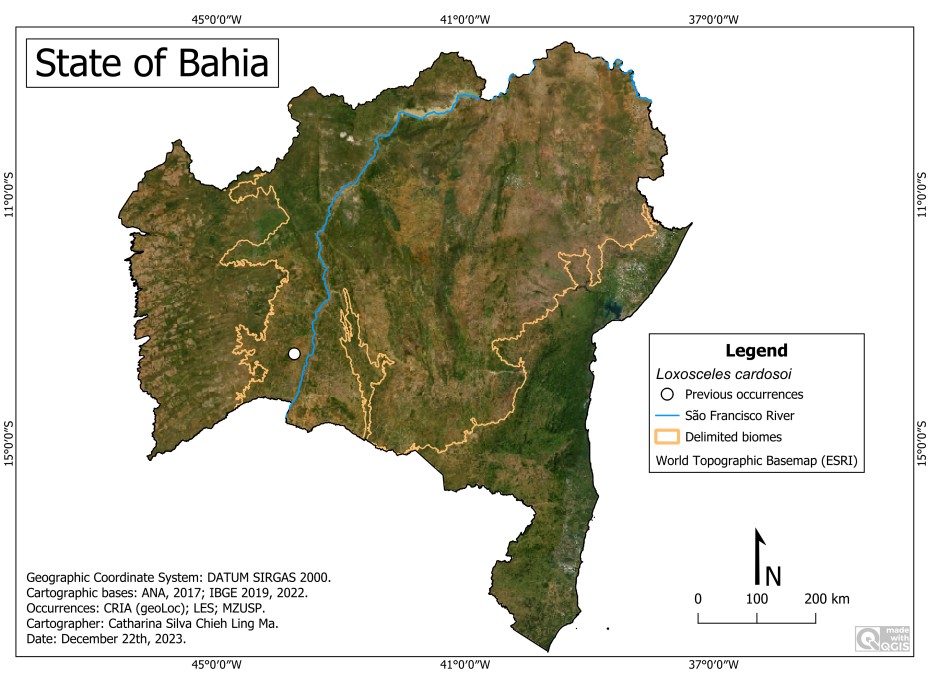

**Figure 12** **Distribution of *Loxosceles cardosoi* in Bahia.** Map data: ESRI Satellite 2023 World Topographic Basemap.

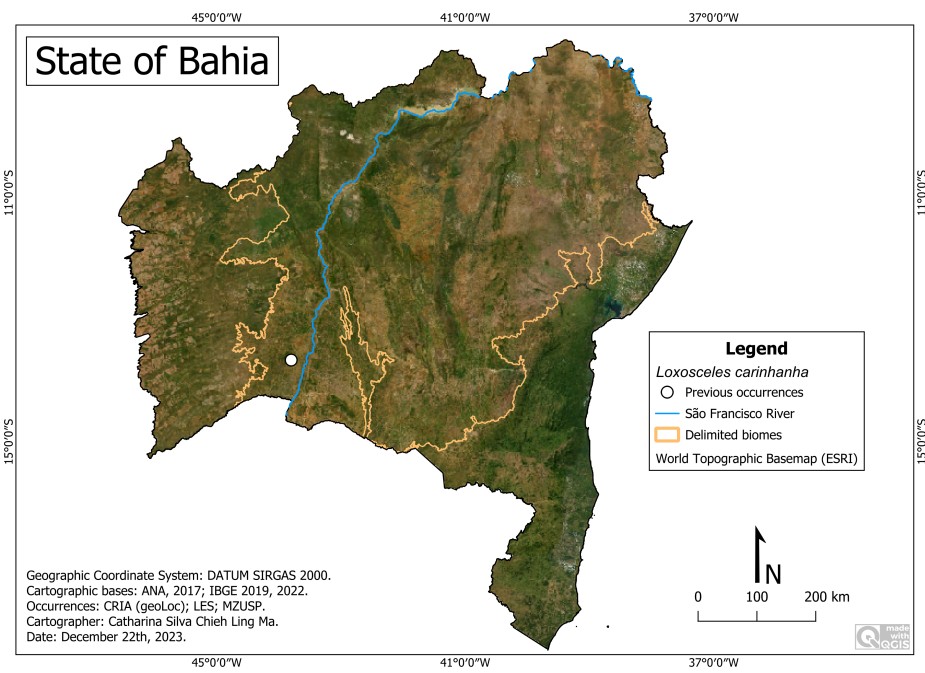

**Figure 13** **Distribution of *Loxosceles carinhanha* in Bahia.** Map data: ESRI Satellite 2023 World Topographic Basemap.

### *Loxosceles chapadensis* Bertani, Fukushima & Nagahama, 2010

**Historical records:** BRAZIL. **Bahia:** *Lençóis* (41°23′23″W, 12°33′48″S); *Palmeiras* (41°33′31″W, 12°31′44″S); *Iraquara*, Fazenda Pratinha (41°32′04″W, 12°21′01″S) (*Bertani, Fukushima & Nagahama, 2010.*)

**New records:** BRAZIL. **Bahia:** *Iraquara*, Gruta da Lapa Doce (41°37′10″W, 12°14′56″S) (UFMG-ARA 12045); Gruta da Torrinha (41°37′10″W, 12°14′56″S) (UFBA-ARA 5612); *Ituaçu,* Gruta da Mangabeira (41°17′47″W, 13°48′48″S) (UFMG-ARA 12095; 15373); *Lençóis,* Gruta do Lapão (41°23′23″W, 12°33′48″S) (UFBA-ARA 3564, 3565, 3612, 3615, 3664, 4168, 4169, 4378, 4381, 4382, 4393, 4394, 4398, 4401, 4408, 4412, 4416, 4417, 4421, 4422, 4503, 4562, 4754, 4755, 4759, 4763, 4768, 4772, 4823, 4900) (UFMG-ARA 12105; 12104; 12103) (IBSP 165648; 165649; 165650; 165651;165652;165653;165654); *Maracás* (40°25′50″W, 13°26′28″S) (UFBA-ARA 4476); *Palmeiras*, Gruta do Riachinho (41°33′31″W, 12°31′44″S) (UFBA-ARA 4491; 4779; 4780).

**Remarks:** Historical records have shown that this species is restricted to the Caatinga, specifically within the Chapada Diamantina National Park (Fig. 14). New records in the municipality of Maracás, close to this Park, show that it inhabits the transition between Caatinga and Atlantic Forest. Records include altitudes of 552 m (Ituaçu) to 964 m (Maracás), average temperature of 21.0 °C (Maracás; Palmeiras) to 22.2 °C (Iraquara) and average precipitation of 637 mm/year (Lençóis; Palmeiras) to 967 (Ituaçu) millimeters/year. We highlight its presence in PARNA-CD (Lençóis), in a place with a recognized tourist influx.

### *Loxosceles karstica* Bertani, von Schimonsky & Gallão, 2018

**Historical records:** BRAZIL. **Bahia:** *Carinhanha*, Gruna do Cocho cave (43°46′W, 13°36′S) (*Bertani et al., 2018*).

**New records:** BRAZIL. **Bahia:** *Caetité,* Bahia Mineração-BAMIN (42°28′29″W, 14°04′10″S) (UFBA-ARA 3609; 3610; 3625; 4607; 4610; 5419); *Maracás* (40°25′50″W, 13°26′28″S) (UFBA-ARA 4770; 4774; 4775; 5102); *Paulo Afonso,* Barragem da Usina Hidrelétrica Luiz Gonzaga (38°12′52″W, 9°24′22″S) (UFBA-ARA 1072).

**Remarks:** The only known record for the state of Bahia indicated that this species had cave-dwelling habits (Fig. 15). The new records indicate its presence in other environments, outside the caves, at altitudes between 243 m (Paulo Afonso) and 964 m (Maracás), average temperature between 21.0 Co (Maracás) and 26.3 Co (Carinhanha) and precipitation average between 367 mm/year (Paulo Afonso) and 885 mm/year (Caetité).

### *Loxosceles similis* Moenkhaus 1898

**Historical records:** BRAZIL. **Bahia** *Pau Brasi l* (39°39′04″W, 15°27′51″S) (*Brazil et al., 2009*).

**New records:** BRAZIL. **Bahia:** *Guanambi* (42°46′52″W, 14°13′25″S) (UFBA-ARA 2673); *Palmeiras*, Vale do Capão (41°33′31″W, 12°31′44″S) (IBSP- 165464); *Pé de Serra*, Fazenda Vanádio de Maracás (39°36′44″W, 11°50′03″S) (UFBA-ARA 2652; 2653); *Ubaíra*, Fazenda Lagoa do Boi (39°40′06.2″W, 13°16′01″S) (UFBA-ARA 2674; 3560; 3561; 3562; 3563).

**Remarks:** The records include locations with altitudes of 169 m (Pau Brasil) to 870 m (Palmeiras), an average temperature of 21.0 °C (Palmeiras) to 25.3 °C (Guanambi), and

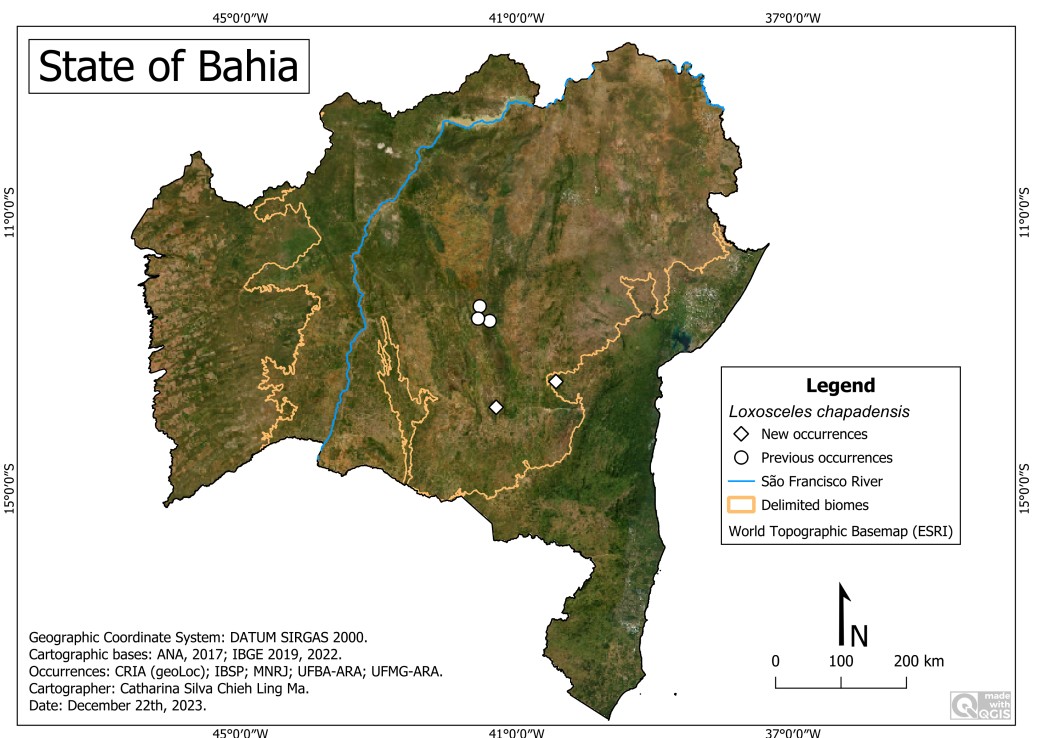

**Figure 14** **Distribution of *Loxosceles chapadensis* in Bahia.** Map data: ESRI Satellite 2023 World Topographic Basemap.

an average rainfall of 491 mm/year (Guanambi) to 863 mm/year (Pau Brasil) (Fig. 16). We highlight its presence in PARNA-CD (Palmeiras), in a place with a recognized tourist influx. Although the record of Pau Brasil was not obtained from either a description article or collections, we chose to include it because it had been previously recorded by the authors (*Brazil et al., 2009*).

### *Loxosceles troglobia* Souza & Ferreira, 2018

**Historical records:** BRAZIL. **Bahia:** *Malhada*, Gruta Tapera D'Água (43°40′58.23″W, 14°31′2.17″S); *Iuiu*, Gruta Taboca (43° 41′10.87″W, 14°35′13.11″S) (*Souza & Ferreira, 2018*).

    **New records**. None

    **Remarks:** Known records show that this species is restricted to caves in the Caatinga (Fig. 17), with altitudes of 433 (Malhada) to 490 m (Iuiu), average temperature of 25.5 °C (Iuiu) to 26.3 °C (Malhada) and average rainfall of 665 (Malhada) to 804 mm/year (Iuiu).

    *Loxosceles* specimens from the MHNBA and arachnidarium (Fig. 18)

## DISCUSSION

Our results indicate the major distribution of sicariid species towards the center and east of the state of Bahia, starting from the east bank of the São Francisco River and with a concentration in the Caatinga (92.8%). This concentration admits the hypothesis of the
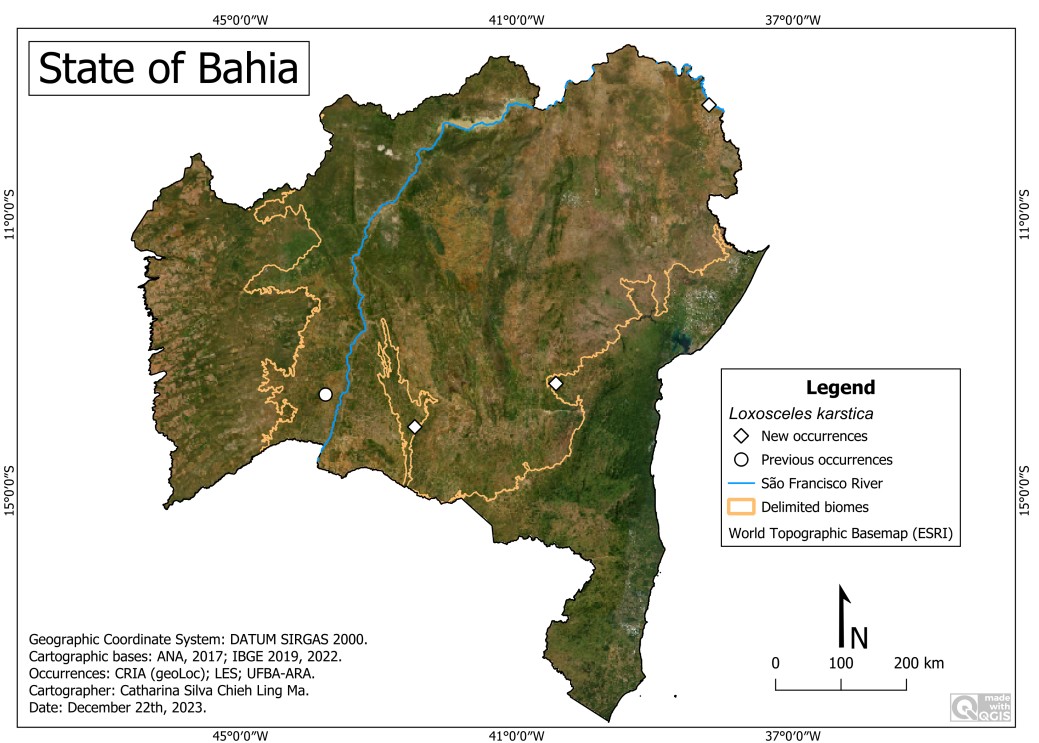

**Figure 15** **Distribution of *Loxosceles karstica* in Bahia.** Map data: ESRI Satellite 2023 World Topographic Basemap.

interrelationship between the species and the characteristics of the biome, whose xerophilic vegetation is predominant and the soil is of sedimentary origin, favoring the establishment of spiders such as *Sicarius*, which have the habit of burying themselves in sandy soil. More than 75% of the species were registered in the east side and about 20% (*Loxosceles boqueirao*, *L. cardosoi*; *L. carinhanha*) exclusively in the west side.

The São Francisco River is an important geographic barrier, especially in the dispersal of small invertebrates. *Da Silva, Pinto-Da-Rocha & De Souza (2015)* consider it an important geographic barrier in the distribution of harvestmen and raise the hypothesis that other taxa with similar biological requirements, with low vagility (such as sicariid spiders), may have patterns of similar endemism or population structure. However, they also consider the possibility that the phytophysiognomic transition is a more important barrier than the river itself. Whether it is one or the other of these barriers, it is necessary to consider the hypothesis of accidental transport, especially considering that the São Francisco River is an important waterway, responsible for the flow of agricultural production from western Bahia (*Ministry of Transport of Brazil, 2018*). Therefore, we have to consider that these spiders can be transported from one area to another through passenger luggage or commercial product packaging.

Certainly, sicariids spiders are not prone to dispersal by balloonism. Ballooning is a well-known dispersal mechanism for small spiders, typically in their first instars, allowing them to transport themselves by air currents, kilometers from their take-off point (*Foelix,*

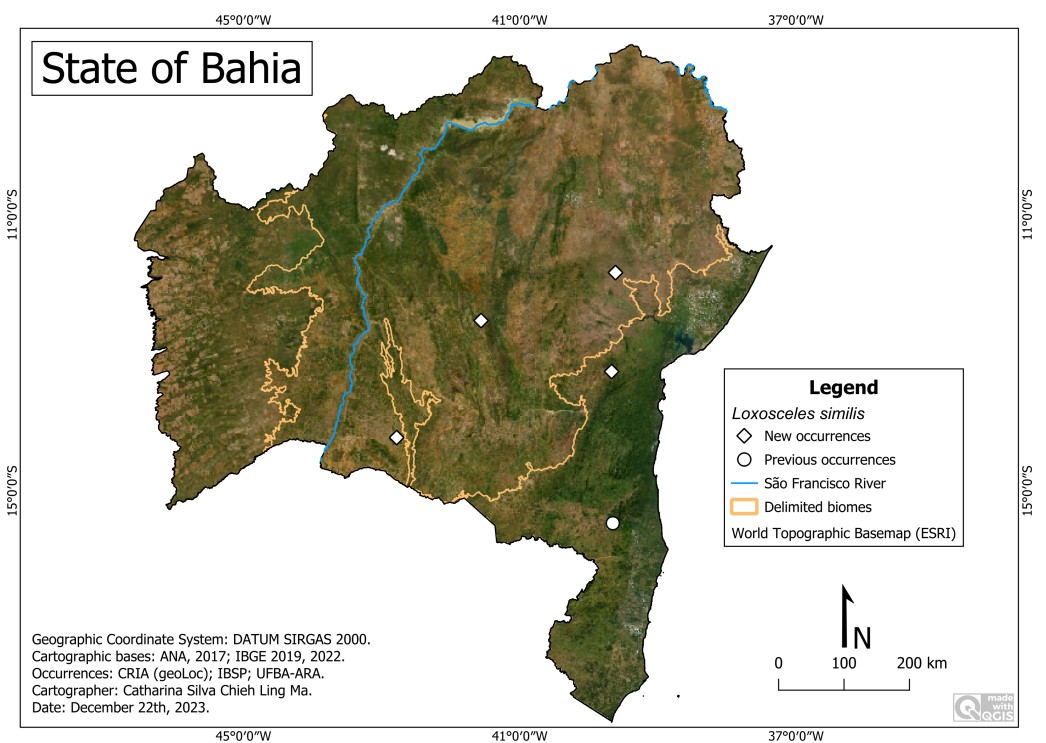

**Figure 16** **Distribution of *Loxosceles similis* in Bahia.** Map data: ESRI Satellite 2023 World Topographic Basemap.

*1996*). This does not appear to be the case with these haplogynae spiders. *Loxosceles* spiders almost depend on humans for long-range transportation (*Vetter, 2008*). Therefore, it is likely that sicariids in Bahia can also occur with a wide distribution in the west bank of the São Francisco River. However, this can only be confirmed through investments in new research in this area.

*Loxosceles* spiders occupy a wide variety of habitats in natural, urban, and domiciliary situations (*Gertsch, 1967*) where they spin irregular webs and generally remain indoors. Many of the species recorded here (*Loxosceles boqueirao, L. cardosoi, L. carinhanha, L. chapadensis, L. karstica,* and *L. troglobia*) came from the interior and from cave entrance areas, or nearby areas, which provide attractive habitats for these nocturnal spiders. However, it is worth mentioning that most records of *Loxosceles* spiders in Brazilian caves are still at a generic taxonomic level (*Trajano, 1987*) which underscores the need for further study in this environment. *Trajano & Bichuette (2010)* report that *Loxosceles* collected in karst areas are those that have probably adapted well to the hypogean environment and are considered troglophiles. Most *Loxosceles* spiders found in caves are troglophiles (facultative cave dwellers) as they can complete their life cycle inside and outside caves, with source populations in epigeal and hypogeal habitats, with gene flow between habitats (*Trajano & Carvalho, 2017*; *Bertani et al., 2024*).

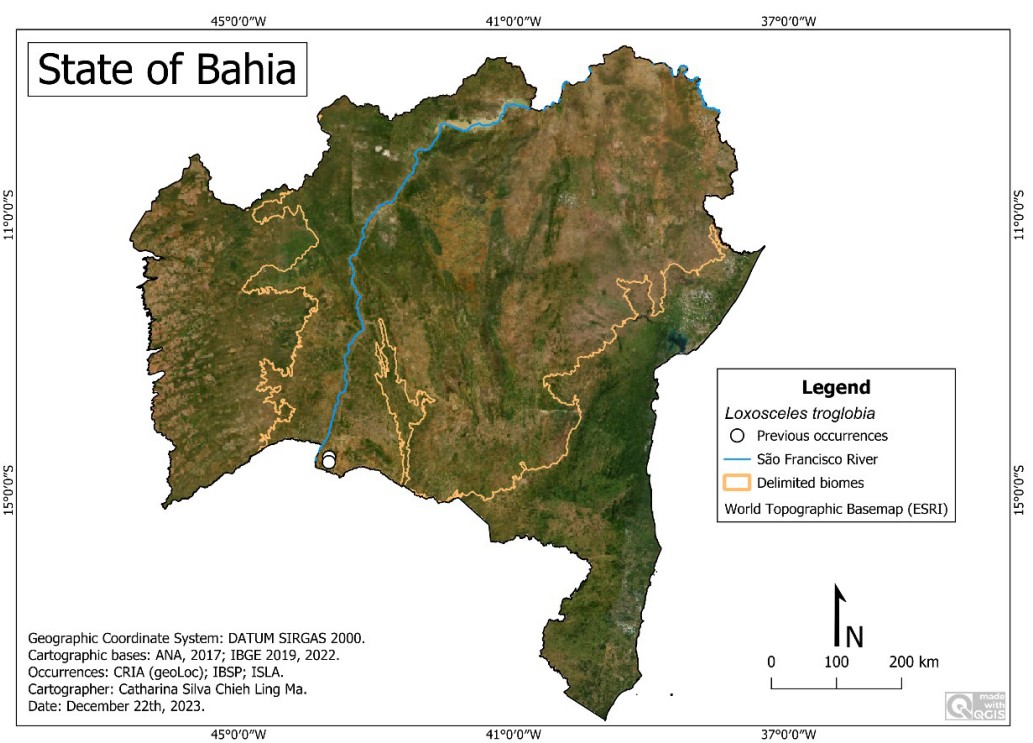

**Figure 17** **Distribution of *Loxosceles troglobia* in Bahia.** Map data: ESRI Satellite 2023 World Topographic Basemap.

*Loxosceles chapadensis* has been recorded inside and outside caves (*Bertani, Fukushima & Nagahama, 2010*; *Andrade-de-Sá, Lira-da-Silva & Brazil, 2023*), what could be an indicator of troglophilia but is not presented in its description.

Troglobite species (population of obligatory and exclusive underground origin) were also recorded, as is the case of *Loxosceles boqueirao* and *L. troglobia* (*Bertani et al., 2024*; *Souza & Ferreira, 2018*)

Some of the recluse spiders *Loxosceles* have become notorious for their toxic venoms capable of injuring humans, reported as loxoscelism. The species that have been related to loxoscelism in Brazil are *Loxosceles intermedia*, *L. laeta,* and *L. gaucho* and most accidents occur indoors, predominantly in the South and Southeast of the country (*Ministry of Health of Brazil, 2011*). None of these species have been recorded in Bahia. Among those listed in this article, only *L. amazonica* has a record of a proven accident, in the state of Ceará (*Lucas, Cardoso & Moraes, 1984*). We emphasize here that all *Loxosceles* accidents reported to date in the State of Bahia have a suggestive diagnosis based on clinical symptoms or proven at the genus level, but do not present proof of the species. It is important to carry out prospective or retrospective studies that relate the clinical case to the species causing the accident.

The Sicariinae are part of a group generally restricted to dry environments (*Magalhães, Brescovit & Santos, 2017*). Although they are found almost exclusively in the Caatinga and

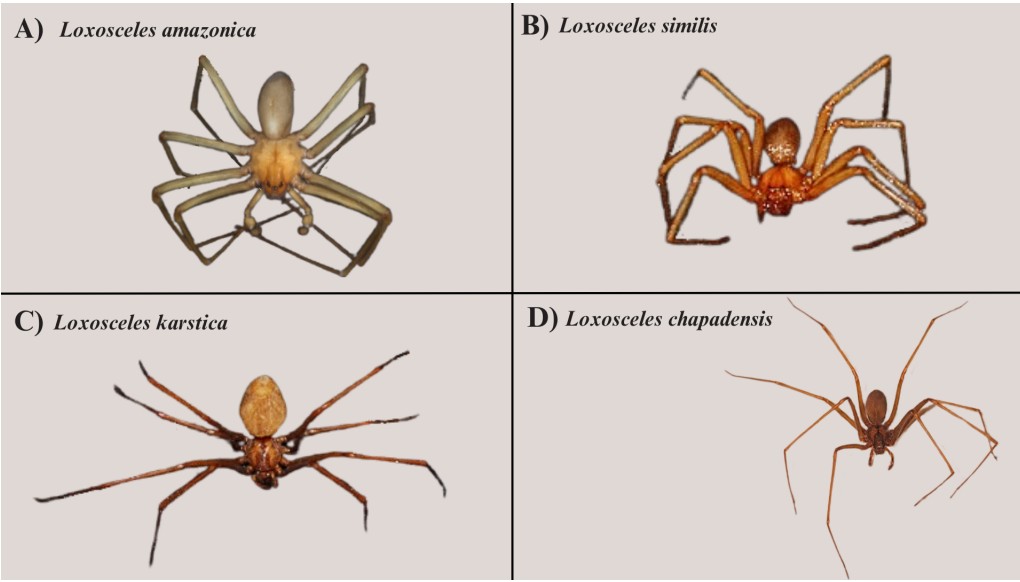

**Figure 18** **Photos of *Loxosceles* from the MHNBA and arachnidarium.** (A) *Loxosceles amazonica* specimen (UFBA-ARA 4189), female from Oeiras, Piauí. (B) *L. similis* specimen (UFBA-ARA 2652), female from Fazenda Vanádio de Maracás, Pé de Serra, Bahia. (C) *L. karstica* specimen (UFBA-ARA 4775), female from Maracás, Bahia. (D) *L. chapadensis* specimen, live female from Iraquara, Bahia (RG: 16164). Photo credit: Júlia Andrade-de-Sá.

are often found in caves, there are isolated records of Restinga and dry forest enclaves in the Cerrado domain (*Magalhães, Brescovit & Santos, 2013*). However, here we present records in the humid forests of the Atlantic Forest of Bahia. This was detected for four species: *Sicarius cariri*, *S. jequitinhonha*, *S. ornatus* and *S. tropicus*. Considering their limited dispersal capacity and their ancestral preference for arid environments, *Magalhães et al. (2019)*, suggest that at more local levels, they may be able to adapt to slightly wetter environments.

## CONCLUDING REMARKS

We consider herein that the universe of data obtained is representative of the region's araneological fauna since they are based on scientific collections that cover a period of 40 years: from 1983 (IBSP) to 2024 (MHNBA). Therefore, we can confirm 14 species of sicariid spiders occurring in the State of Bahia, grouped into two subfamilies: Sicariinae (six *Sicarius*) and Loxoscelinae (eight *Loxosceles*), distributed in 48 municipalities, which corresponds to approximately 11.5% of the 417 municipalities in the State. Among these species, six (*L. boqueirao*, *L. cardosoi*, *L. carinhanha*, *L. chapadensis*, *L. troglobia*, *S. saci*) have not been recorded in other Brazilian states yet and may be strong candidates for an endemicity status.

Our results also indicate that the Chapada Diamantina ecoregion is an important support for the diversity of karstic sicariid spider species in the State of Bahia, as well as the Serra do Ramalho region.

The striking difference in species composition in the two regions separated by the São Francisco River suggests that future studies should be carried out to evaluate additional variables that determine the structure of these spiders' communities, especially considering their low vagility and recluse behavior. Factors that contribute to the structuring of local and regional spider assemblages, such as climate, vegetation complexity, and dispersal capacity, must be evaluated.

## ACKNOWLEDGEMENTS

We thank Professor Elizabeth Neves for allowing us to use the photographic magnifying glass and Catharina Ma for supporting the production of the maps.

### Funding

This work was financially supported by Fundação de Amparo à Pesquisa do Estado da Bahia (FAPESB) scholarship to Júlia Andrade-de-Sá (Proc. 084.0508.2021.0003001-19); Conselho Nacional de Desenvolvimento Científico e Tecnológico (CNPq grant PQ 316413/2021-6) to Rejâne M. Lira-da-Silva; and Conselho Nacional de Desenvolvimento Científico e Tecnológico (CNPq grant PQ 303903/2019-8 and FAPESP grant 2022/12588-1 to Antonio D. Brescovit. The funders had no role in study design, data collection and analysis, decision to publish, or preparation of the manuscript.

### Grant Disclosures

The following grant information was disclosed by the authors:
Fundação de Amparo à Pesquisa do Estado da Bahia (FAPESB): Proc. 084.0508.2021.0003001-19.
Conselho Nacional de Desenvolvimento Científico e Tecnológico: CNPq grant PQ 316413/2021-6, PQ 303903/2019-8 and FAPESP grant 2022/12588-1.

### Competing Interests

The authors declare there are no competing interests.

### Author Contributions

- Júlia Andrade-de-Sá conceived and designed the experiments, performed the experiments, analyzed the data, prepared figures and/or tables, and approved the final draft.
- Tania Kobler Brazil conceived and designed the experiments, analyzed the data, authored or reviewed drafts of the article, and approved the final draft.
- Rejâne Maria Lira-da-Silva analyzed the data, authored or reviewed drafts of the article, and approved the final draft.
- Antonio Domingos Brescovit analyzed the data, authored or reviewed drafts of the article, and approved the final draft.

## Data Availability

The fourteen distribution maps pinpointing the records of each sicariid species in Bahia, one overall distribution map, and one map of the research area; photos of *Loxosceles* and of *Sicarius* are available in the Supplementary Files.

## Supplemental Information

Supplemental information for this article can be found online at http://dx.doi.org/10.7717/peerj.17942#supplemental-information.

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
