# Peer review of "The sicariid spiders in the state of Bahia, Brazil (Arachnida: Araneae)"

_PeerJ, doi:10.7717/peerj.17942_

## Round 0.1 · original submission · Major Revisions

The manuscript was evaluated by three expert reviewers, all of whom were enthusiastic about the study. However, they did point out a number of issues that should be addressed in a revised version of the manuscript. In particular, I agree that these are simply distribution maps, rather than any sort of data "analysis". Because you do have geographic data, there are certainly more comprehensive analytical ways of modeling species distributions (where might these spiders actually be located); I will leave that to your discretion (i.e., whether you want to actually conduct Maxent type analyses). If you retain these as distribution maps only, I would suggest either an overview map showing all of the spider species' distribution plotted, and/or combining a few of the individual maps; I think this would be easier on the read and allow for a more comparative type of analysis.

Please do provide a detailed response to each of the reviews when you submit your revised manuscript.

Reviewer 1 ·

Basic reporting

The English is ok, however the title must to be changes to: The Sicariidae spiders (Arachnida: Araneae) in the State of Bahia, Brazil. The original title said: (Araneae; Arachnida)... so the correct is first the Class, after that the Order. The Introduction needs additional information of previous papers published about dermonecrotic accidents, there is a bunch of information about this topic and here is missing, please provide some updated papers to be cited. The maps are ok, with good quality, however, the photographs of the specimens are too dark, please to take picture in lighter background.

Experimental design

The research is interesting because they use from several records from several years for many species, as the authors said: "The work did not start with an initial hypothesis, the main question was: who are the Sicariidae species occurring in Bahia, and where can we find them? It is crucial to address these inquiries because all these spiders possess the enzyme sphingomyelinase D in their venom, responsible for the dermonecrotic activity observed in loxoscelism accidents"... However, altought both genera Sicarius and Loxosceles have the Sphingomyenilase D, the accident in Sicarius are really rare because the ecology of these species living in sand into dry habitats, whereas the MOST importat cases of dermonecrotic accidents are by Loxosceles, so, this is not a real argument to support the assumption, which has no hypothesis as such. What other evidence the authors have to support this assumption?

About the Rigorous investigation performed using high thecnics or analyses, this contribution do not have any analysis, the distribution records into a map altuough provide important information about the distribution, they did not implemented any posterior analyses such as species distribution modeling or/and species niche modeling, which might provided additional information. They use the Average annual temperature and rainfall and Biomes and Physiognomies, however, this is only to delimitate them into de maps distribution but not for analize the distribution records as bioclimatic variables or using an algorith such as Maxent. They have in some species more than 3 records, which are necessary to implement species distribution modeling, but none analysis there is into the manuscript, to map distribution records using QGIS is not an analyisi, is onlt a tool to show those records into a map.

Validity of the findings

The authors mentioned that the São Francisco River is an important geographic barrier, however The São Francisco River is an important geographic barrier, however, only using a modeling of distribution species would be tested, I know that this is not the goal of the contribution, but only species registration records fails to test this geographic feature as a important barrer... The auhtos should include into the discussion some work with Species distribition modeling to have additional data to discuss this part (see Saupe et al 2019).

Also about the discussion of the Loxosceles from karstic caves, there is an important research about ecological difference between Loxosceles tenochtitlan and L. misteca from Mexico, this last one use to be into kartic caves, and the lenght of the legs were longer than L. tenochtitlan, a common species in central Mexico, so the ecology and microhabitat are important to drive the diversification of this genus (see Valdez-Mondragón et al. 2019) with the description of L. tenochtitlan. This paper can provide additional discussion for the species from caves and their diversification. More discussion is necessary about this assumption.

Additional comments

Altought the author into the Discussion section talk about the clinical syntomps and loxoscelism, it is necessary and updated additional information about this topic, see paper of Vetter to discuss about the clinicak syntoms, the most of them as Richard Vitter suggest are clinical misidentifications, maybe emphasize this point, please see the Vetter´s book published in 2015: The brown recluse spider, chapter: Medical misdiagnoses.

Annotated reviews are not available for download in order to protect the identity of reviewers who chose to remain anonymous.

Reviewer 2 ·

Basic reporting

This manuscript focuses on providing new and updated records for spiders of the genus Loxosceles and Sicarius from Bahia, Brazil. The article is well written, and the maps are of good quality. I understand the main objective of the manuscript is to provide new records and distribution maps for the family, however, I believe authors can improve their manuscript by including some extra information and possible extra analyses that can give a boost to their manuscript and data.

Experimental design

Some of the results presented in "Concluding remarks" section can be converted into plots that can help visualize the species composition differences, for example, abundancy and diversity plots per municipality or biome, this way will be easier to see the differences in number of species or abundancy. For example, a heatmap can be plotted over the study area denoting species abundance to emphasize the differences produced by the São Francisco River. If authors already have raster files for all this species, this can be used as basis for simple analysis of diversity with species richness and/or abundancy, including some basic diversity indexes per region, biome, municipality, etc.

Finally, inclusion of some analyses directly associating presence/absence data (rasters) with environmental layers can be done to see any habitat preference/difference, this last suggestion involves performing more analysis but these can be very interesting for maybe unveiling some interesting patterns on habitat preference by genera and or location.

Validity of the findings

no comment

Reviewer 3 ·

Basic reporting

acceptable

Experimental design

acceptable

Validity of the findings

acceptable

Additional comments

PeerJ review

This is a straight forward contribution to the sicariid literaure. However is PeerJ a taxonomic journal? The few papers I have seen in PeerJ are aimed at a more general audience. This manuscript seems more appropriate for Zootaxa,

In the abstract, it gives direction as to left or right. That depends if you are looking north or south. Use east and west instead.

Sicariidae is a noun and should be started with a capital S and never used as an adjective. "sicariidae spiders" is incorrect. The adjective form is "sicariid spiders" all lower case.

The venom component Sphingomyelinase D is now known as Phospholipases D.

In the discussion, it says "injuring man". This is antiquated. Change to "injuring humans"

I assume PeerJ is online only. If also a print version, then there are too many figures with minimal information and should be combined.

---

## Round 0.2 · accepted · Accept

The authors have done an acceptable job addressing most of the criticisms raised in the previous review. I agree that the paper is more descriptive than analytical and thus am OK moving forward without additional analyses. Note that the reviewer points out a number of instances where "Sicariidae spiders" is improperly termed - I trust that will be remedied in the final copy edit version.

Reviewer 3 ·

Basic reporting

basic reporting is fine

Experimental design

It is a straight forward paper in my opinion

Validity of the findings

there isn't much to criticize here either

Additional comments

1- there are still instances of "Sicariidae spiders" in the title and abstract which is wrong. It should either be "sicariid spiders" or "spiders of the Sicariidae family"

2-line 237: change "most" to "greatest"